# Ceramic Matrix Composites Obtained by the Reactive Sintering of Boron Carbide with Intermetallic Compounds from the Ti-Si System

**DOI:** 10.3390/ma15238657

**Published:** 2022-12-05

**Authors:** Dawid Kozień, Izabella Czekaj, Patrycja Gancarz, Magdalena Ziąbka, Wojciech Wieczorek, Katarzyna Pasiut, Dariusz Zientara, Zbigniew Pędzich

**Affiliations:** Faculty of Materials Science and Ceramics, AGH University of Science and Technology, 30 Mickiewicz Av., 30-059 Kraków, Poland

**Keywords:** composites, UHTCs (ultra high-temperature ceramics), boron carbide, intermetallic, TiSi_2_, Ti_5_Si_3_

## Abstract

In this study, we investigated the effect of adding two different intermetallics, Ti_5_Si_3_ and TiSi_2_, for the preparation of TiB_2_-SiC-B_4_C composites. As part of the research, stoichiometric composites consisting only of two phases TiB_2_ and SiC were obtained. The TiB_2_-SiC-B_4_C composites were prepared via pressureless sintering. The presence of the phases in the sintered composites was confirmed using X-ray diffraction and scanning electron microscopy. The SEM-EDS examination revealed that the TiB_2_ and SiC phases were formed during the composite process synthesis and were distributed homogeneously in the B_4_C matrix. The obtained results allowed us to usually exceed 2000 °C and the use of specialized equipment for firing, that is, vacuum or protective atmosphere furnaces as well as control and measurement equipment. Such an approach generates high costs that are decisive for the economics of the technological processes. In the case of our compositions, it is possible to lower the temperature to 1650 °C. The TiB_2_-SiC-B_4_C composites were classified as UHTCs.

## 1. Introduction

Ceramic composites, based on boron carbide (B_4_C) are very promising candidate materials for structural applications, due to their superior mechanical, chemical, and thermal properties [1,2]. Boron carbide has a very low density (2.52 g/cm^3^), compared with other popular ultra-high-temperature ceramics (UHTCs), such as TiC (4.93 g/cm^3^) [3], ZrB_2_ (6.08 g/cm^3^) [4], and AlN (3.26 g/cm^3^) [5,6]. Additionally, boron carbide has a high hardness (up to 40 GPa) and a high melting point (2450 °C). The combined properties give it wide-ranging applications in various industries, mostly in the military as light-weight armor, and bulletproof materials, and also in nuclear applications, such as control rods, thanks to its high resistance against radioactive emissions, which makes B_4_C an excellent neutron absorber [7]. Moreover, boron carbide shows several limitations, e.g., low fracture toughness (3–4 MPa·m^0.5^) and a poor sinter ability which is mainly caused by the high content of the strong covalent bond B-C [6,7]. This directly leads to a low value of the self-diffusivity coefficient. In addition, an oxide layer of B_2_O_3_ is located on the grain surface of the B_4_C powder, which acts as a contaminant and is difficult to remove even at elevated temperatures during the sintering process. The pressureless sintering technique enables obtaining a relative density of 94.5% at 2230 °C; however, high values can be achieved by using the hot-pressing technique at 2150 °C [8].

Many studies have shown that introducing various phases into boron carbide to form a multiphase composite creates a satisfactory way to improve the compaction and fracture toughness of the material [6]. Second-phase particles, such as TiB_2_ [9] or SiC [10], are specifically desired not only because of their low density and high hardness, but also because they improve the sinter ability, fracture toughness, and total hardness of the composite [1,11,12]. Boron carbide is often used as an additive to promote the densification in UHTC composites, such as ZrB_2_ [13]. B_4_C functions by removing the surface oxides (B_2_O_3_ and ZrO_2_) present on the surfaces of the starting powder particles through solid-state sintering [4,13]. In addition, the hot-pressing temperature required for the densification can be reduced to as low as 1600 °C using a boron carbide additive that promotes the liquid-phase formation [13]. In general, the use of additives to improve the sintering and toughness of B_4_C ceramics can be summarized into two categories: solid-state and liquid-phase sintering [1]. Composites fabricated via transient liquid-phase sintering can be widely used because they preserve the advantages of both solid- and liquid-state sintering. This sintering process is more energy-efficient than pressureless sintering in the solid-state because of the sintering additives that promote the liquid-state sintering process to occur at lower temperatures. The liquid phase proceeding from the additives reacts with solid-state particles in the material and forms new phases, which appear in a multi-particulate microstructure [14]. Despite these advantages, liquid-phase sintering is not entirely adequate because the remaining intergranular phase irretrievably impairs the hardness and fire resistance of the resulting B_4_C composites [15].

The liquid phase-sintering procedure consists of adding a reactive sintering additive with a melting point explicitly below the sintering temperature of the process, or on the occurring reactions with the matrix when the product of the reaction gives a low melting point reactive product. Intriguingly, in the specific case of boron carbide, its reaction with the sintering additive will automatically produce other borides and carbides, which exhibit similar, although a lower than B_4_C, hardness. This method of sintering has come to be applied to boron carbide very recently, many types of research performed so far show that the obtained composites, based on boron carbide and the sintering additive exhibit better properties, such as hardness [16,17], fracture toughness [16,17], Young’s modulus [16], and abrasion [15] than the bulk boron carbide ceramics fabricated under always interchangeable conditions [14].

The SiC-TiB_2_-TiC composites can be an excellent composite material for cutting tool applications, have specific properties, such as a lightweight and high mechanical, thermal, electrical, and tribological properties [18,19,20]. The currently known methods of synthesizing SiC-TiB_2_-TiC composites are characterized by a high synthesis temperature above 2000 °C and, depending on the phase composition, may require a synthesis temperature of over 2200 °C [21,22,23]. In the case of the results presented in the article, it is possible to reduce the synthesis temperature to 1650 °C, which significantly reduces the cost of synthesis

In the present work, the highly densified B_4_C-SiC-TiB_2_ composites were obtained by reactive pressureless sintering with B_4_C and Ti_5_Si_3_, and TiSi_2,_ separately, as raw materials. In this study, both powder substances were selected as additives to produce composites with a wide range of applications and excellent properties The respective hardening, strengthening, and toughening mechanisms of the resulting ceramic composites are discussed, in exploring the relationships between microstructure and mechanical properties.

## 2. Thermodynamic Calculations

Based on the compounds described, two reactions are considered for each additive that may occur in a given system. Using thermodynamic data from various sources [2,24,25,26] and by establishing stoichiometric coefficients in the reactions under consideration, the ΔG of these reactions was calculated.
(1)$${\mathrm{Ti}}_{5}{\mathrm{Si}}_{3}+B_{4}C+5C\to 2{\mathrm{TiB}}_{2}+3\mathrm{TiC}+3\mathrm{SiC}$$
(2)$$2{\mathrm{Ti}}_{5}{\mathrm{Si}}_{3}+5B_{4}C+C\to 10{\mathrm{TiB}}_{2}+6\mathrm{SiC}$$
(3)$$3{\mathrm{TiSi}}_{2}+B_{4}C+6C\to 2{\mathrm{TiB}}_{2}+\mathrm{TiC}+6\mathrm{SiC}$$
(4)$$2{\mathrm{TiSi}}_{2}+B_{4}C+3C\to 2{\mathrm{TiB}}_{2}+4\mathrm{SiC}$$

It was initially assumed that the dominant reaction for the Ti_5_Si_3_ system will be the one in which the products are titanium boride (TiB_2_), silicon carbide (SiC), and titanium carbide (TiC), while the formation of the first two products was assumed for the TiSi_2_ system. During the theoretical considerations, to conclude the dominant reaction in the considered systems, the calculations were made and a graph of the dependence of ΔG on the temperature was made for reactions 1–4 (Figure 1).
(5)Reaction (1) ΔG=−877180+77.51·T
(6)Reaction (2) ΔG=−2774300+209.8·T
(7)Reaction (3) ΔG=−771380−35.16·T
(8)Reaction (4) ΔG=−627580−17.44·T

Figure 1 shows that the most energetically beneficial reaction is reaction (2) because it has the lowest ∆G value in the temperature range from 298K to 2698K. The value of the free enthalpy is information about the stability of a given compound, the lower it is, the more durable the chemical compound is. Based on the above calculations, the reaction taking place in the B_4_C and Ti_5_Si_3_ systems was assumed to be reaction (2), which produces titanium boride (TiB_2_) and silicon carbide (SiC).

Based on the analysis of the TiSi_2_ and B_4_C systems, it can be concluded that reaction (3) is energetically more favorable, in which both TiB_2_, SiC, and TiC are formed because it has a lower value of ΔG in the entire temperature range considered. Considering the value of the free enthalpy ΔG over the entire temperature range, the differences between reactions (3) and (4) are small. Because titanium has a greater affinity for boron than for carbon, it was assumed that the second reaction would take place in the system under consideration, consistent with Equation (4). This is confirmed by the results of the experimental studies, later in the study.

## 3. Materials and Methods

### 3.1. Materials

Commercially available B_4_C powder (B_4_C—Grade HD 07 produced by H. C. STARCK, Goslar, Germany) was used as the base material. The starting powder had a purity of 99% (1% of soot as sintering additive), a specific surface of 6–9 m^2^/g, and the medium size of the particle was about 2 μm and the molar mass of the powder was equal to 55.26 g/mol, as reported by the manufacturers. Titanium silicide powders (TiSi_2_ and Ti_5_Si_3_) were synthesized using self-propagating high-temperature synthesis (SHS). The phases were fabricated by the exothermic reaction between the Ti and Si powders with proportions of Ti_5_Si_3_ and TiSi_2_ phases. The substrates were homogenized and after that were separated from the grinding media and transferred to the reactor chamber. The SHS synthesis was initiated by resistive heating under a protective atmosphere. The obtained products were ground using an Abich mortar. The full procedure for the synthesis of the titanium silicide powders is described in [27] and also in the patent application number P.437232 [28].

### 3.2. Procedures

Prior to proceeding with the practical operations, the appropriate calculations were performed according to the 2nd and 4th reactions considered in the thermodynamics section. The amounts of B_4_C powder and soot (99% B_4_C and 1% soot in commercial powder) required to maintain the proper reaction stoichiometry were calculated.

Individual compositions were prepared using commercial boron carbide with the addition of 5%, 10%, 15%, and 20% by weight TiSi_2_ and Ti_5_Si_3_. The compositions were also converted into 5 g of the sample and the corresponding amounts of B_4_C were weighed, successively for 5% additives 0.25 g, for 10% additives 0.50 g, for 15% additives 0.75 g, and 20% additives 1 g of powder. In addition, the right amount of B_4_C powder itself was weighed and a “control” sample was prepared, as well as samples with stoichiometric compositions of TiSi_2_ and Ti_5_Si_3_. They were then manually homogenized in a mortar in an isopropanol medium and dried. The homogenized powders were collected, and lozenges with a diameter of 10 mm were made using a manual press; they were also subjected to isostatic pressing at a pressure of 200 MPa using a National Forge press. The resulting moldings were subjected to free sintering in a dilatometer with the simultaneous recording of the changes in the linear dimensions of the sample. The sintering process was carried out in an argon atmosphere (Ar), and the measurements were carried out from 750 °C to 2150 °C, and the temperature progress was 10 °C/min. The obtained composites were cooled together with the device. Following the sintering, the sinters were prepared using a grinder-polisher from Struers RotoPol-22. Grinding discs were used with the following gradations: 80, 220, 500, 600, 1200, and 2000 using the diamond paste, and at the very end a polishing canvas was used.

The relative density, open porosity, and water absorption of the samples were measured, according to Archimedes’s method. The phase compositions were determined via X-ray diffraction (XRD Panalytical/Philips X’Pert Pro MD Diffractometer) with a Cu-Kα1 radiation. Young’s modulus was designated using the measurement of the longitudinal wave passage speed through the material (EPOCH 650 Olympus flaw detector, Hamburg, Germany). The hardness was determined using a Vickers hardness testing machine (Future-Tech FM-700 hardness tester, Kawasaki-City, Japan), performing three tests for each sample with a 9.81 N load. The method of measuring the Palmqvist crack length in the Vickers hardness test was used to calculate the fracture toughness KIc, based on the Nihara–Moreno–Hasselmann formula. Four tests were performed, and 9.81 N force was used, respectively. The microstructure characterization and elemental mapping were performed by scanning electron microscopy (ThermoFisher Scientific Phenom XL SEM with CeB_6_ source) with an EDS spectrometer and SED detector. The morphology of the particles was analyzed by scanning electron microscopy (Nova NanoSEM 200, FEI Company, Eindhoven, the Netherlands) with an energy-dispersive X-ray spectrometry system. The analysis was performed at an accelerated voltage (10–15 kV) and under high vacuum conditions with a BSE detector.

## 4. Results

### 4.1. Characteristics of the Sintering Process

The Figure 2a–d shows the obtained results as linear shrinkage curves of the pure commercial boron carbide and composites with determined compositions. All the sintering curves were obtained with a 10 °C min^−1^ heating rate. The shrinkage of commercial boron carbide without an additive sample (Figure 2a) started at 1600 °C. For the addition of 5% TiSi_2_, the sintering process began at approximately 1600 °C whereas the remaining additives obtained the sintering temperatures at a range of 1700–1800 °C (Figure 2b). The highest dimensional shrinkage for this additive was registered for the 5% content of TiSi_2_ in the composite. The sintering improvement was achieved for samples with an additive Ti_5_Si_3_ of 5% and 15% (Figure 2c), as evidenced by the reduced sintering temperature and the greater shrinkage of the sample, compared to the sintering of the commercial B_4_C sample. The greatest change in composite dimensions during the sintering was recorded for 15% of the Ti_5_Si_3_ addition. The stochiometric addition of TiSi_2_ (Figure 2d) increased the sintering temperature to 1700 °C. In the composite with the stochiometric addition of Ti_5_Si_3_ (Figure 2e), the lowest temperature at which the sintering process started was around 1420 °C. A slight dimensional shrinkage was obtained for the sample with a stochiometric amount of TiSi_2_, and an inconsiderable dimensional increase for the stochiometric amount of Ti_5_Si_3_.

### 4.2. Density Measurement and Porosity

The results of the apparent density and porosity measurements for the pressureless-sintered composites at 2250 °C and HP-ed composites are shown in Table 1. Regardless of the number of additives introduced, the densities obtained for isostatically pre-pressed composites were higher than those for pressureless-sintered composites. Moreover, in the case of the hot-pressed composites, the total porosity was lower. The isostatic pressing improves the densification and reduces the porosity of the materials. In most compositions, a similar density was observed when compared to that of the pure commercial B_4_C.

### 4.3. Mechanical Properties

The results of Young’s modulus measurements are shown in Figure 3. For both additives, a regular dependence of Young’s modulus on the increasing percentage of the additive is observed. Better results were obtained for specimens with the TiSi_2_ addition. The highest values of Young’s modulus were recorded, and the increase in this value was related to the increasing amount of the additive. The highest value recorded for 69.635% TiSi_2_ was 162 GPa and for 69.344% Ti_5_Si_3_ was 143 GPa. The lowest value recorded for 5% of both additives, was 113 GPa and 97 GPa, respectively. Compared to the commercial boron carbide, Young’s modulus values for the individual composites are lower than those reported in the literature [29].

Hardness measurements were conducted on the hot-pressed samples. The obtained results for all compositions are presented in Figure 4, which shows the hardness of the composite with the changing percentage of the specified additive. Both of the presented composites increase their Vickers hardness with the increasing additive content, which shows the same trends with the change of the apparent density in Table 1. The hardness of all samples, sintered by hot-pressing process (HP) at 1850 °C, is understated. The sintered sample of the commercial boron carbide can obtain a hardness up to 35 GPa [30]. A tendency of increase in the hardness with the increasing additive content is observed. Samples with 5–20% of both additives tend to be characterized by similar hardness values to each other. For the TiSi_2_ additive, the obtained values of hardness are slightly higher than the values received for the samples with the Ti_5_Si_3_ additive. The maximum hardness is obtained in the samples with the stochiometric additive content, which are 16.4 GPa for TiSi_2_ and 11.7 GPa for Ti_5_Si_3_, respectively.

Fracture toughness measurements (K_Ic_) are shown in Figure 5. The trend of this property is different for the specimens depending on the used additive and its content. According to the literature, fracture toughness for pure boron carbide oscillates between 2.9–3.7 MPa·m^1/2^ [31]. The lowest values of the fracture toughness K_Ic_ were recorded for the composites with TiSi_2_, for the 5% addition, K_Ic_ was 1.53 MPa·m^1/2^. The highest fracture toughness was also recorded for the addition of TiSi_2_, the maximum K_Ic_ value was achieved for stochiometric composition with this additive, which was 2.67 MPa·m^1/2^. The values of the fracture toughness K_Ic_ for the addition of Ti_5_Si_3,_ reached the values from 2.30 MPa·m^1/2^ for 5% Ti_5_Si_3_, up to 2.31 MPa·m^1/2^ for the stochiometric composition with this additive, which was, and so it remains a not very good result, compared to the pure boron carbide.

The X-ray diffraction analysis of the composites showed that for an increasing amount of intermetallic (TiSi_2_ and Ti_5_Si_3_) additives, the percentage of the new phase, i.e., the titanium boride (TiB_2_), increased (Table 2). The quantitative content of graphite in the sinter with a 15% additive is much greater than in the sinter with boron carbide without the use of the additive. This is because after the complete conversion of Ti_5_Si3 with B_4_C, TiB_2_ and SiC phases are formed in the system and the excess carbon (C) from the B_4_C decomposition remains. The reaction takes place this way because titanium (Ti) has a greater affinity for boron (B), while silicon (Si) reacts with carbon (C) first. The reaction proceeded with 100% efficiency because we do not observe the presence of the Ti_5_Si_3_ or TiSi_2_ phases. When increasing the amount of intermetallics, in samples from 5–20% of the additive, we do not observe unreacted intermetallic residues, and on the diffraction patterns of the commercial boron carbide sinter with a stoichiometric amount of the additive used, a partial conversion of titanium silicide (Ti_5_Si_3_ or TiSi_2_ depending on the additive used) was recorded in the tested system. The percentage of the remaining phase in both cases was below 3% (Figure 6a,b), which confirms the thermodynamic calculations made with the assumption of the course of the reaction for both additives. In the phase composition of the previously observed phases, i.e., titanium boride (TiB_2_) and silicon carbide (SiC), the presence of a new phase, i.e., titanium carbide (TiC), was also observed, with the simultaneous lack of graphite in the sample. Earlier thermodynamic considerations ruled out the appearance of titanium carbide (TiC), but its presence was registered due to inhomogeneities in the sample. Titanium silica (Ti_5_Si_3_) in the observed place completely reacted with boron carbide (B_4_C), forming titanium boride (TiB_2_) and silicon carbide (SiC). An excess amount of titanium (Ti) remained in the system, which began to react with the carbon resulting from the decomposition of boron carbide (B_4_C), resulting in the formation of a titanium carbide (TiC) phase.

### 4.4. Microstructure Observation

The SEM micrographs and elemental analysis of the B_4_C and TiSi_2_ or Ti_5_Si_3_ additives in the composites synthesized at 1850 °C under hot-isostatic pressing (HIP) are given in Figure 7. The microstructure of the B_4_C composite with the stoichiometric addition of TiSi_2_ (Figure 7a) and the microstructure of the B_4_C composite with stochiometric addition of Ti_5_Si_3_ (Figure 7b) as well as the distribution maps of the elements are presented. The pseudo-color maps show the distribution of elements: boron, carbon, titanium, and silicon in the microstructure of the composite. The silicon and titanium precipitation is observed: the greater the molar fraction of silicone or titanium in the composite, the greater the areas of silicone or titanium occurrence. The elemental maps facilitated the analysis of the new phases containing titanium and boron formed during sintering.

The SEM images showing the microstructures of composites with TiSi_2_ and Ti_5_Si_3_ additives are shown in Figure 8 and Figure 9, respectively. Both additives act similarly, the greater the molar fraction of the elements incorporated into the composites, the greater the occurred areas of the new phases. Samples with the Ti_5_Si_3_ additive tend to have less porosity than the samples with the other additives. Samples with the stochiometric addition of TiSi_2_ and Ti_5_Si_3_ are more homogenized than the composites with a smaller percentage of additives.

## 5. Discussion

The present article focuses on the synthesis of the TiB_2_-SiC-B_4_C composites by the pressureless sintering. The process was conducted in a dilatometer that allowed to track the shrinkage kinetics. By analyzing the sinterability of the materials, it can be stated that the amount and type of the added intermetallic phase have an impact on lowering the sintering temperature. The sintering curves of the composite materials with a stoichiometric share of Ti-Si additives show how the progress of the sintering process changes, at which temperature the material intensifies the most, and what the sintering process unfolds as a function of time. The SEM analysis allowed us to analyze the microstructure of the obtained composites and compare the morphology of the obtained composites, free sintering leads to the consolidation and distribution of individual phases in the composite, the degree of their distribution depends however on the amount of the additive. The greater the amount of additive used for sintering, the greater the density of the final material, as well as its porosity and water absorption. Compared to the starting materials, the composites show higher values of these properties for the addition of TiSi_2_. Moreover, the addition of Ti_5_Si_3_ causes an increase in density, but the porosity and water absorption decrease as its amount increases. The increase of intermetallic phase content caused a significant increase of hardness. As a result, the sintering resulted in a greater number of phases with a higher hardness (from 2.4 to 14.88 GPa), resulting in a greater total hardness of the final material. The selection of an appropriate intermetallic material allows to control composition of the composites by increasing the TiB_2_ or SiC phase content in the final product composition or, as in the case of stoichiometric samples, it allows for obtaining composites consisting only of the TiB_2_-SiC phases with different weight proportions. By using an addition in the form of Ti_5_Si_3_, we obtained a smaller amount of B_4_C phase in the composites, but also more in the TiB_2_ phase, which is also characterized by a high hardness. Therefore, when comparing these two additives, it can be concluded that the use of Ti_5_Si_3_ gives a better effect, because in total we received a greater number of phases with a higher hardness, which translates into a general increase in the hardness of the material.

The applied intermetallics (Ti_5_Si_3_, TiSi_2_) and the sintering technique also improve the mechanical properties of the composites due to the TiB_2_ and SiC phase formation. These two phases probably significantly improve the B_4_C–TiB_2_–SiC properties. It is surmised that the rapid densification of the composite sintered between 1600 and 1650 °C may arise from the accelerated mass transport between the solid phases and the formed eutectic liquid phase which can fill pores. The obtained composites can be a material for cutting tools (both in the form of coatings and composites for fittings), and their great advantage over the currently used cutting tools is a high thermal resistance due to the presence of the TiB_2_ phase.

## 6. Conclusions

The conclusions drawn from the results can be summarized as follows:Due to the presence of a liquid phase from the intermetallic materials, it is possible to lower the synthesis temperature from 2000 °C (currently used when using other precursors) to 1650 °C. During the sintering, the boron carbide stoichiometry converts from B_4_C to B_13_C_2_.When TiSi_2_ was used as an additive, an overall increase in the sintering temperature was observed, excluding the smallest amount of additive. The opposite is true for the addition of Ti_5_Si_3_; in this case, the effect is a decrease in the sintering temperature.A chemical reaction taking place during the sintering is very effective. Almost 99% of the initial phases decompose and allow the formation of new TiB_2_ and SiC phases, which are well densified at relatively low temperatures. The TiC phase is formed only when no boron is present in the system during sintering; hence, in the case of the addition of Ti_5_Si_3_, when there is a significant amount of Ti in the system and a lack of boron, a small amount of the TiC phase is formed.The microstructure and mechanical properties of the B_4_C–TiB_2_–SiC ceramic composites can be effectively tuned by regulating the combinations of the composition of the starting precursors used, in particular through the use of various intermetals from the Ti-Si system. In the composites B_4_C–TiB_2_–SiC carbon residue is observed in the phase composition, formed during the synthesis in the composites with the use of traditional phase compositions obtained. The presence of carbon in the microstructure significantly reduces its mechanical strength. In the case of using boron carbide and intermetals from the Ti-Si system and the affinity of carbon for silicon, and titanium for boron, there is no carbon left in the system, which weakens the structure of the obtained composites.Both additives tend to be characterized by similar hardness values to each other. A higher hardness obtained for the addition of intermetallic TiSi_2,_ in relation to the addition of Ti_5_Si_3_ are associated with obtaining a larger amount of the SiC phase, which has a higher hardness than the TiB_2_ phase. The maximum hardness is obtained in the samples with stochiometric additive content, which are 16.4 GPa for TiSi_2_ and 11.7 GPa for Ti_5_Si_3_, respectively.

## 7. Patents

Patent application to the Patent Office of the Republic of Poland entitled “The method of obtaining composite highly refractory made of boron carbide and an intermetallic compound from the Ti-Si system” with the number: P.437232 [WIPO ST 10/CPL437232].

## Figures and Tables

**Figure 1 materials-15-08657-f001:**
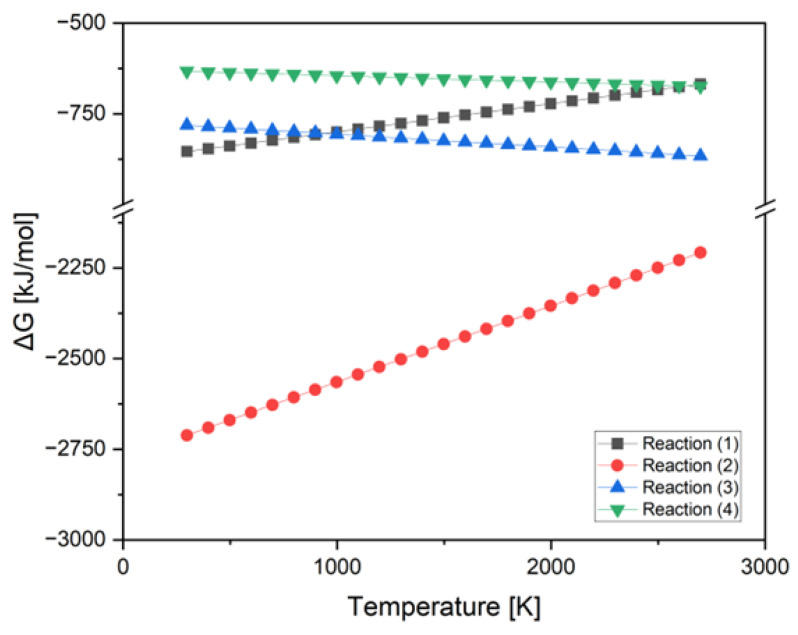
ΔG of reactions 1–4 as a function of the temperature.

**Figure 2 materials-15-08657-f002:**
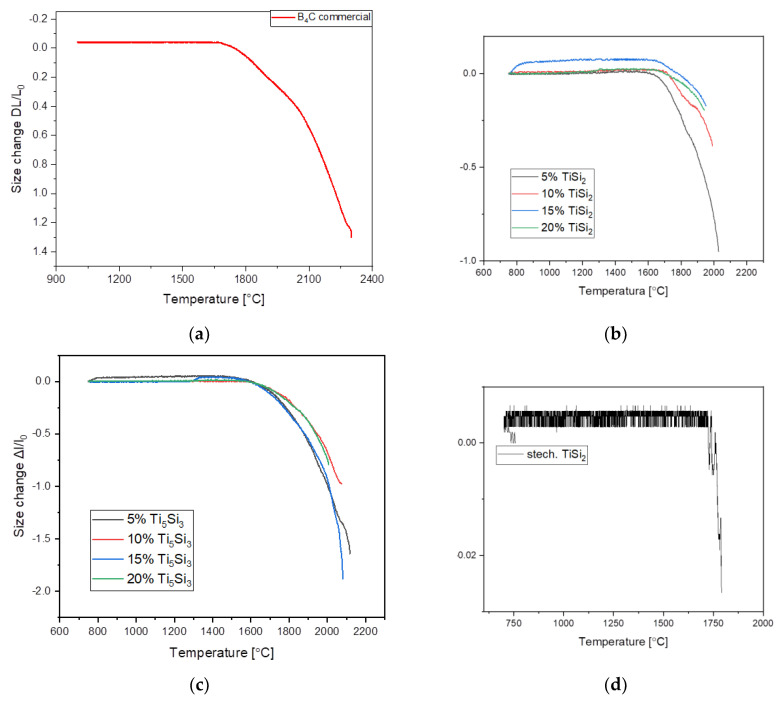
Dilatometric curves of the composite sintering process at a temperature of 2250 °C. (**a**) commercial B_4_C, (**b**) B_4_C +%TiSi_2_, (**c**) B_4_C +%Ti_5_Si_3,_ (**d**) B_4_C + stochiometric additive TiSi_2_, (**e**) B_4_C + stochiometric additive Ti_5_Si_3_.

**Figure 3 materials-15-08657-f003:**
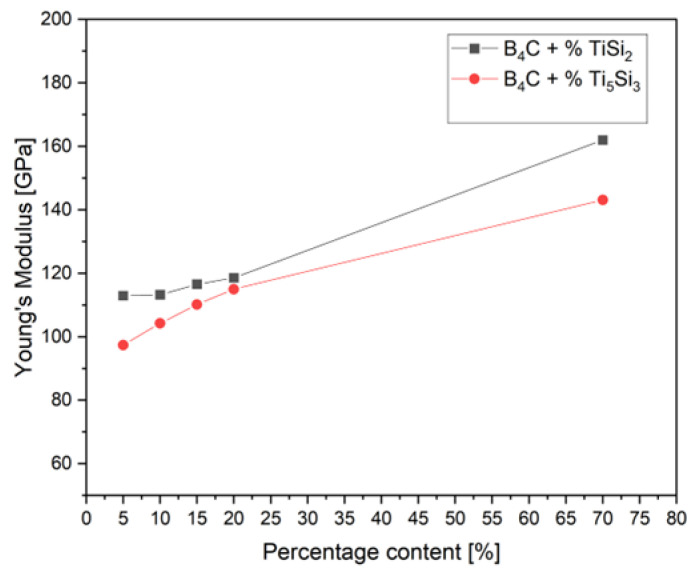
Young’s modulus of the composites as a function of the percentage of various additives.

**Figure 4 materials-15-08657-f004:**
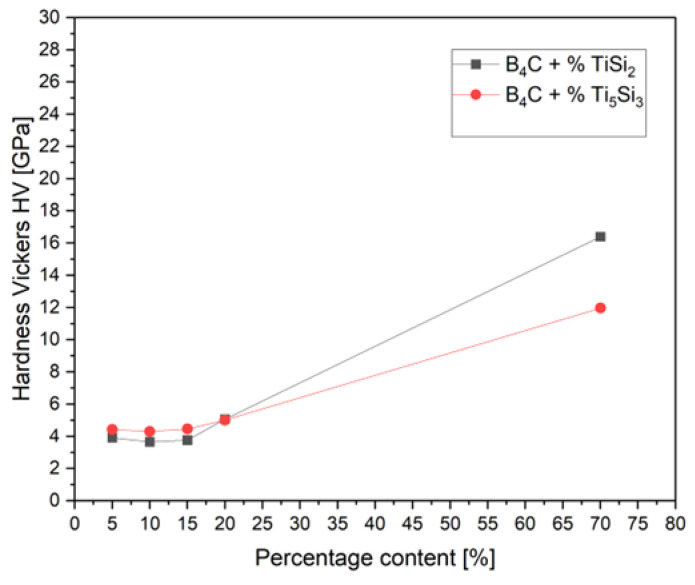
Hardness of the composites as a function of the percentage of various additives.

**Figure 5 materials-15-08657-f005:**
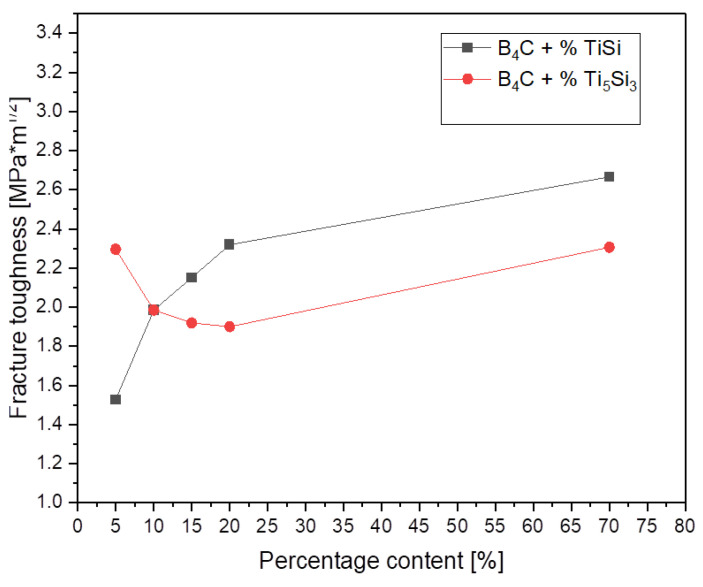
Fracture toughness of the composites as a function of the percentage of various additives.

**Figure 6 materials-15-08657-f006:**
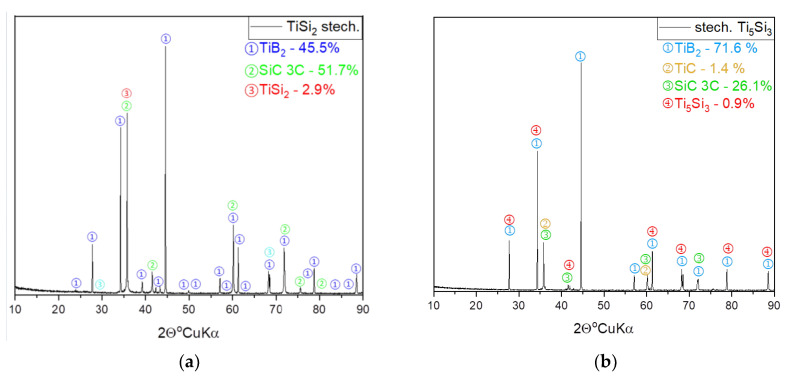
Diffractograms for the composites obtained from the stoichiometric samples in which the addition of the intermetallics was: (**a**) TiSi_2_, (**b**) Ti_5_Si_3_.

**Figure 7 materials-15-08657-f007:**
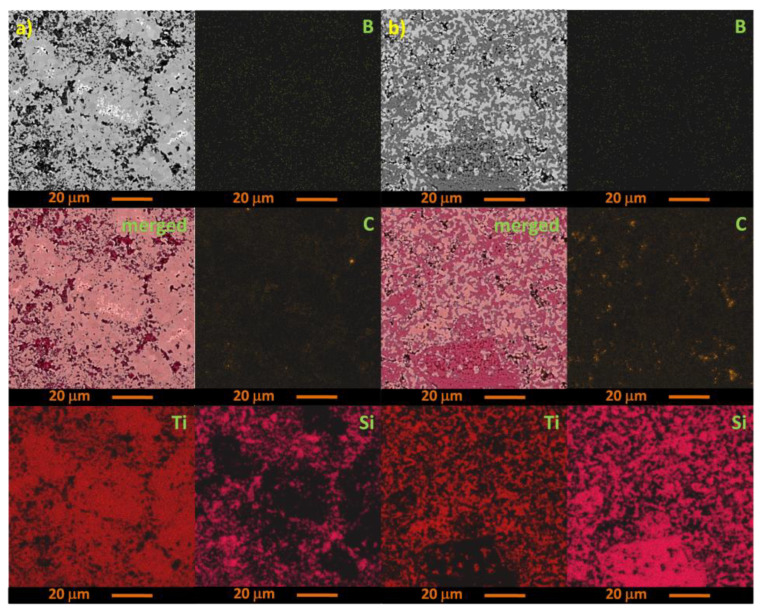
SEM elemental maps for B, C, Ti, and Si showing the microstructures of the composites: (**a**) stochiometric addition of TiSi_2_ (**b**) stochiometric addition of Ti_5_Si_3_.

**Figure 8 materials-15-08657-f008:**
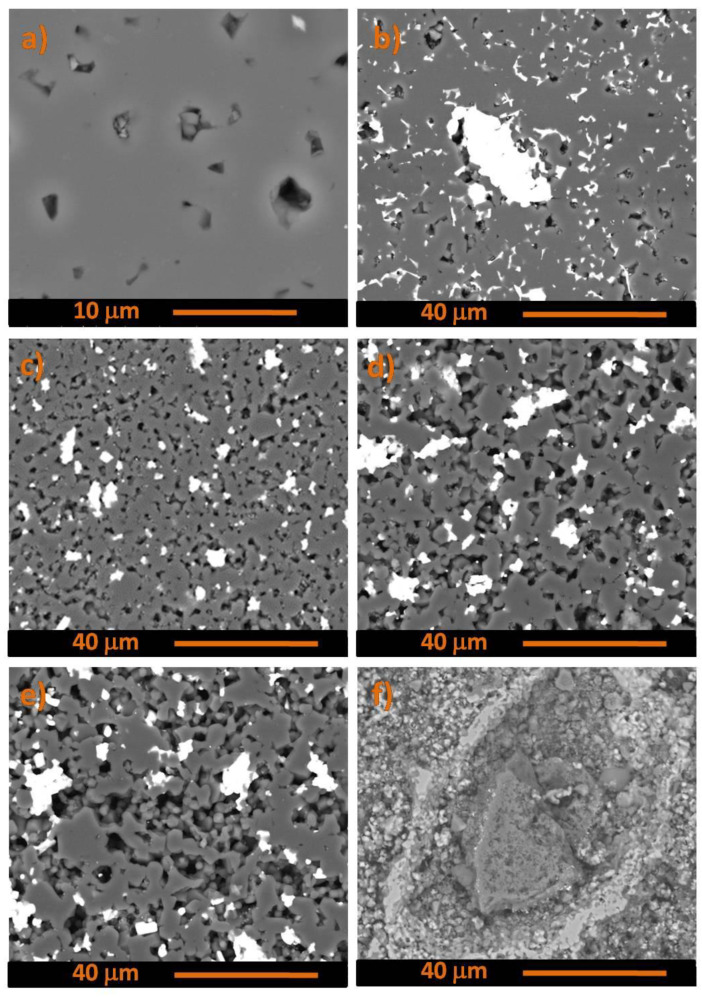
The SEM images showing the microstructures of the composites B_4_C: (**a**) 0% TiSi_2_, (**b**) 5% TiSi_2_, (**c**) 10% TiSi_2_, (**d**) 15% TiSi_2_, (**e**) 20% TiSi_2_, (**f**) stochiometric addition of TiSi_2_.

**Figure 9 materials-15-08657-f009:**
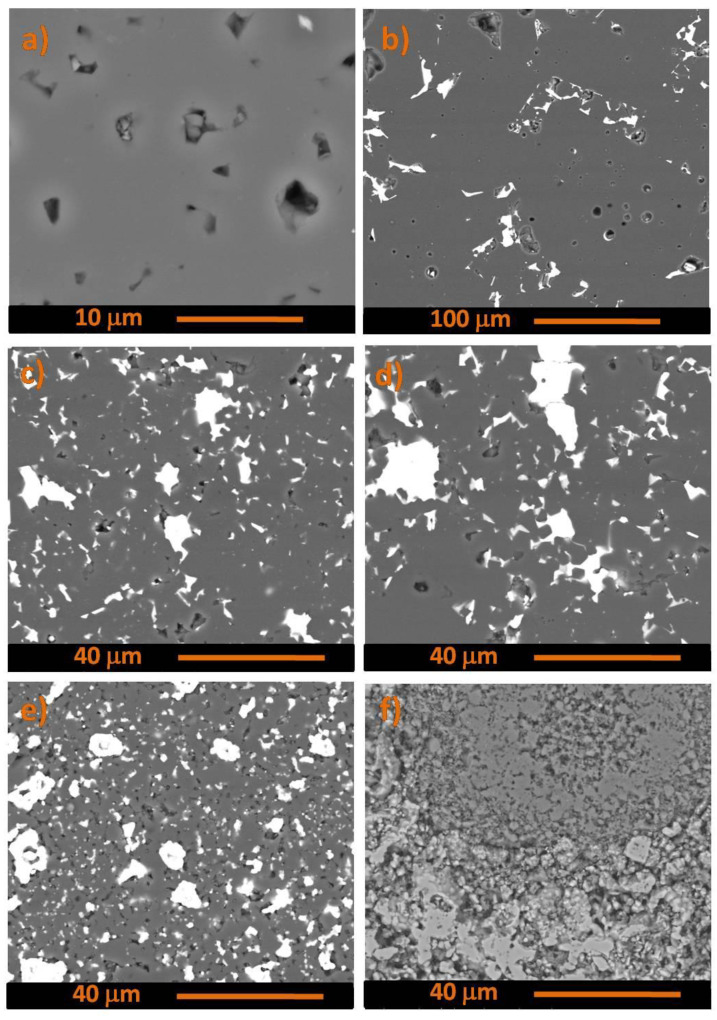
The SEM images showing the microstructures of the composites B_4_C: (**a**) 0% Ti_5_Si_3_, (**b**) 5% Ti_5_Si_3_, (**c**) 10% Ti_5_Si_3_, (**d**) 15% Ti_5_Si_3_, (**e**) 20% Ti_5_Si_3_, (**f**) stochiometric addition of Ti_5_Si_3_.

**Table 1 materials-15-08657-t001:** Apparent density, porosity calculated values, and the phase composition of B_4_C with TiSi_2_ and Ti_5_Si_3_ fabricated at 2250 °C and hot-pressing (HP) at 1850 °C.

Sample	Apparent Density [g/cm^3^]		Porosity [%]	
	Sintering at 2250 °C	Hot-Pressing at 1850 °C	Sintering at 2250 °C	Hot-Pressing at 1850 °C
B_4_C_com_	2.37	2.45	1.65	0.05
B_4_C + 5% TiSi_2_	2.12	2.31	3.41	1.46
B_4_C + 10% TiSi_2_	2.26	2.52	17.83	6.63
B_4_C + 15% TiSi_2_	2.29	2.51	23.81	7.02
B_4_C + 20% TiSi_2_	2.35	2.59	25.59	4.58
B_4_C + 5% Ti_5_Si_3_	2.28	2.46	13.33	1.00
B_4_C + 10% Ti_5_Si_3_	2.33	2.51	13.56	0.21
B_4_C + 15% Ti_5_Si_3_	2.47	2.54	16.96	0.53
B_4_C + 20% Ti_5_Si_3_	2.58	2.63	17.86	2.06
B_4_C + 69.635% TiSi_2_	2.51	3.86	33.08	2.40
B_4_C + 69.344% Ti_5_Si_3_	3.46	3.70	35.92	5.86

**Table 2 materials-15-08657-t002:** The phase composition of the composites with Ti_5_Si_3_ and TiSi_2_ after sintering.

Precursor	Composition [%]			
	B_13_C_2_	Graphite [C]	TiB_2_	SiC
B_4_C commercial	99.0	1.0	-	-
B_4_C + 5% TiSi_2_	93.3	0.4	4.7	1.7
B_4_C + 10% TiSi_2_	92.9	0.1	5.1	1.9
B_4_C + 15% TiSi_2_	86.0	0.3	7.6	6.1
B_4_C + 20% TiSi_2_	78.0	-	13.0	9.0
B_4_C + 5% Ti_5_Si_3_	86.4	1.5	11.0	1.1
B_4_C + 10% Ti_5_Si_3_	76.9	0.2	18.5	4.3
B_4_C + 15% Ti_5_Si_3_	77.3	6.6	13.4	2.6
B_4_C + 20% Ti_5_Si_3_	78.6	0.2	20.5	0.8
B_4_C + 69.635% TiSi_2_ *	-	-	45.5	51.7
B_4_C + 69.344% Ti_5_Si_3_ *	-	-	71.6	26.1

* Additive residue after sintering 2.9% TiSi_2_, 0.9% Ti_5_Si_3_ and 1.4% TiC in the stochiometric composite with Ti_5_Si_3_.

## Data Availability

Not applicable.

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
