# Peer review of "Ceramic Matrix Composites Obtained by the Reactive Sintering of Boron Carbide with Intermetallic Compounds from the Ti-Si System"

_materials, 2022, doi:10.3390/ma15238657_

Round 1
Reviewer 1 Report
The paper “Low-temperature method of obtaining dense boron carbide matrix composites utylizing intermetallic compounds from the Ti-Si system in the starting composition” is devoted to preparation and investigation of the TiB2-SiC-B4C composites. Self-propagating high-temperature synthesis and isostatic pressing were applied for the composites sintering. XRD, SEM, EDX techniques were used for the samples characterization. The topic of this paper is critically actual especially in aerospace, nuclear and military industry. The data are reliable and do not cause much doubt. Nevertheless, there are several points before the paper can be published. I hope that authors after major revisions can improve the paper and can publish it in Materials.
1. I think the authors uploaded the working version of the manuscript with the regime of the comments in MS Word.
2. The Introduction part must be improved with relevant literature about other alternative techniques and I suggest to use the following reference (see and discuss:
https://doi.org/10.3390/nano12101642;
https://doi.org/10.1016/j.jeurceramsoc.2019.04.001;
https://doi.org/10.3390/ma15051946).
3. More practical applications should be added to the manuscript.
4. “The phase compositions were determined via X-ray diffraction (XRD Panalytical/Philips X'Pert Pro MD Diffractometer) with Cu-Kα1 radia-tion.” – I didn’t found the XRD results. Please explain it.
5. Conclusion part is too short, please improve it.
6. Language – there are some insufficient typos and mistakes in the text. Please revise it.
Reviewer 2 Report
The current paper reports the synthesis of boron carbide matrix composites at low temperature from Ti-Si system. The major critic of this paper is that, the results were not presented in a proper way and the paper is incomplete. There are comments in the paper, which is totally unacceptable. It seems to be a draft version of the manuscript rather than a final copy. The authors claims they have carried out XRD, though there is not a single XRD spectrum in the manuscript. Some addition experimental details/data are requires as mentioned in the detailed comment section. Based on my assessment, I must reject this paper.
The other comments are as follows:
1. The title may be concise and revised for better clarity. Currently it bit too long.
2. Line 22-26: Those sentences may be avoided in the abstract section.
3. The abstract must be re-written in a focused way. There is no need to add experimental details in the abstract. It will contain the keys findings, in a concise way.
4. Line 35: Include the full form of UHTC before start suing the abbreviation.
5. The last paragraph of the introduction section, should be the aim of the present work, not the process/methodology. The authors should read other papers to find out the writing style of a journal paper!
6. Line 93: All the ref. related to the ‘various sources’ must be included to confirm result reproducibility.
7. Line 182-184: ‘This section may be divided by subheadings…...’—This is not acceptable. The authors should check their work before submitting. It seems the work is incomplete!
8. Fig. 2a and b may be combined together for ease of comparison.
9. Fig. 2 must be presented in better way. In current form it is all misleading and confusing.
10. Section 4.3: How does the Young’s modulus calculated? Included this is the experimental section.
11. Fig. 3 in incomplete! What was the percentage of TiSi2??
12. Same remarks for Fig. 4 and 5.
13. Table 2: How does the phase composition was measured?
14. There is no XRD pattern in manuscript as claimed by the authors in Line 165.
15. The discussion is not convincing. Needs addition SEM/TEM work to support the discussion.
16. The conclusion section will start with a brief overview of the work. The authors should read other papers to find out the writing style of a journal paper!
17. Section 6: Why there is a comment on the side?
Reviewer 3 Report
The paper presents some interesting results of effect of addition of Ti5Si3 and TiSi2 intermetallics on structure and property of dense boron carbide matrix composites, the following questions need to be revised before publication:
1 Why did it change from B4C to B13C2? What is the difference between B4C and B13C2?
2 The XRD results should be described with graphs to provide more information.
Reviewer 4 Report
Dear authors,
You did an interesting work but the presentation of it does not satisfy completely. My remarks are as follows:
Please try to simplify the title of manuscript.
Please describe how you performed the thermodynamic calculations in Section 2.
Please describe procedures of cooling after sintering in Section 3.2.
Please state the time of grinding for each gradation.
Please state the operating conditions for diffractometer and time of indentation for hardness measurements.
In Tables line spacing should be decreased.
In discussion you should to confront your obtained results with results from other researchers dealing with similar issue.
You should improve Conclusion by adding conclusions for analysed properties.
Best regards
Round 2
Reviewer 1 Report
Now the paper can be accepted in present form.
Author Response
Thank you very much for all the comments that helped to improve the article.
Reviewer 2 Report
1. Fig. 7: The scale bars are not visible. Needs to be edited like other SEM images.
2. In conclusion section, it should be between 3-5 point and should be revised accordingly.
Author Response
Rev. 1 As suggested by the reviewer, the figure has been corrected. We hope it is acceptable in its current form
Rev. 2 As suggested by the reviewer, the conclusions are contained in 5 point